# Human Intestinal Tissue Explant Exposure to Silver Nanoparticles Reveals Sex Dependent Alterations in Inflammatory Responses and Epithelial Cell Permeability

**DOI:** 10.3390/ijms22010009

**Published:** 2020-12-22

**Authors:** Kuppan Gokulan, Katherine Williams, Sarah Orr, Sangeeta Khare

**Affiliations:** National Center for Toxicological Research, Division of Microbiology, US Food and Drug Administration, 3900 NCTR Rd, Jefferson, AR 72079, USA; kuppan.gokulan@fda.hhs.gov (K.G.); kgmwilliams8@gmail.com (K.W.); seorr@ncsu.edu (S.O.)

**Keywords:** ex vivo intestinal model, human intestine, permeability, immune status, intestinal mucosa, silver nanoparticles

## Abstract

Consumer products manufactured with antimicrobial silver nanoparticles (AgNPs) may affect the gastrointestinal (GI) system. The human GI-tract is complex and there are physiological and anatomical differences between human and animal models that limit comparisons between species. Thus, assessment of AgNP toxicity on the human GI-tract may require tools that allow for the examination of subtle changes in inflammatory markers and indicators of epithelial perturbation. Fresh tissues were excised from the GI-tract of human male and female subjects to evaluate the effects of AgNPs on the GI-system. The purpose of this study was to perform an assessment on the ability of the ex vivo model to evaluate changes in levels of pro-/anti-inflammatory cytokines/chemokines and mRNA expression of intestinal permeability related genes induced by AgNPs in ileal tissues. The ex vivo model preserved the structural and biological functions of the in-situ organ. Analysis of cytokine expression data indicated that intestinal tissue of male and female subjects responded differently to AgNP treatment, with male samples showing significantly elevated Granulocyte-macrophage colony-stimulating factor (GM-CSF) after treatment with 10 nm and 20 nm AgNPs for 2 h and significantly elevated RANTES after treatment with 20 nm AgNPs for 24 h. In contrast, tissues of female showed no significant effects of AgNP treatment at 2 h and significantly decreased RANTES (20 nm), TNF-α (10 nm), and IFN-γ (10 nm) at 24 h. Smaller size AgNPs (10 nm) perturbed more permeability-related genes in samples of male subjects, than in samples from female subjects. In contrast, exposure to 20 nm AgNPs resulted in upregulation of a greater number of genes in female-derived samples (36 genes) than in male-derived samples (8 genes). The ex vivo tissue model can distinguish sex dependent effects of AgNP and could serve as a translational non-animal model to assess the impacts of xenobiotics on human intestinal mucosa.

## 1. Introduction

The increasing availability of commercial products containing nanomaterials, particularly silver nanoparticles (AgNPs), has amplified the need to ensure that such products are safe for consumer use. AgNPs can have considerable activity against bacteria, fungi, and even some viruses [1,2,3,4,5,6,7] making them attractive to companies interested in manufacturing products with food safety or sanitary benefits. A variety of consumer products now contain AgNPs, including food contact materials [8,9,10], clothing and personal care products [11,12] dietary supplements [13], and water filters [14,15]. Additionally, many studies have shown that simulated use of AgNP-containing consumer products can result in AgNP release from these products into the environment [8,9,11,12,16]. For items intended for use in food preparation or for oral administration, this could lead to gastrointestinal tract (GIT) exposure to AgNPs. This is especially of concern to the public, because research studies have demonstrated that AgNPs could negatively impact the gut-associated immune response [17], cytotoxicity [18], inflammation and oxidative stress response [19,20,21], and intestinal barrier function [22]. There are also concerns regarding the effects of nanoparticle exposure on the gut microbiota and maintenance of intestinal homeostasis. Additionally, many studies have examined the overt toxicity of AgNPs on different GI models, but few studies have looked at more subtle impacts, such as increased inflammation, initiation of an immune response, and/or alterations in the intestinal permeability. An increase in inflammation can affect barrier integrity, leading to greater absorption of potentially harmful substances from the intestinal lumen.

Numerous approaches have been utilized to examine nanoparticle effects on the GIT, including in vitro cell culture models and in vivo studies in model animals. In vitro cell culture approaches are advantageous because they are generally lower in cost and provide greater control of exposure conditions for evaluating the effects of test compounds/materials. However, the gastrointestinal epithelium is a complex organ that performs many vital functions and is composed of a variety of separate, but complimentary, cell types. The cell culture approach has several significant limitations in simulating the gastrointestinal environment, such as replicating the essential functions of the mucin layer and associated microbiome [23]. To overcome some of these aspects, studies are conducted by addition of external mucus to form a layer at the apical surface of the cells or employing co-cultures systems that include goblet-like cells or the transformation of some of the cells into M cells [24,25]. Another common strategy is the in vivo approach, where nanomaterials are administered to living animals. This has the advantage of examining nanoparticle effects in a more biologically relevant environment and allows the performance of empirically-determined toxicology testing. However, such studies are almost solely limited to model animals because of ethical concerns surrounding administration of nanoparticles to human subjects. Animals and humans also have physiological differences that suggest that data derived from toxicological studies using animals may not always be applicable to human safety [23]. Researchers have observed considerably different histology patterns when comparing similar tissues excised from animals and humans [26]. Finally, ethical consideration regarding the use of animals in research is a driving motivation for finding alternatives to animal-based experimentation.

For all reasons mentioned above, non-animal models for use in toxicology and safety testing are becoming increasingly desirable, especially for models of epithelial barriers. Furthermore, several researchers have attempted an ex vivo approach, in which tissues excised from human or animal subjects are handled under specific conditions that temporarily preserve their physiological properties and biological function [26]. The term ex vivo is also applied to organoids created from cells that are derived from fresh tissue [27,28,29]. Recently, organoid cultures have been developed and utilized to model intestinal stem cell differentiation and development during disease states [28]. Additionally, organoids may be used in studies involving intestinal tumor initiation, metastatic progression, and therapy response of common GI cancers [30]. These procedures require additional manipulation of excised intestinal tissue and the protocols used are often laboratory-specific. Another emerging model in intestinal mucosa layer research is the “gut-on-a-chip” system, which is used to address interactions of test materials with the GI tract [31,32,33]. Apart from the cost, it has limited utility to investigate effects on subpopulations within a cohort (e.g., sex-dependent responses). It is important to note that this manuscript uses the term ex vivo solely to refer to the utilization of freshly excised human intestinal tissue. Unlike organoid, 3D cell culture, or gut-on-a-chip models, the ex vivo model used in this study involves minimal manipulation by the researcher. Several different approaches with tissue explant models have been tested, primarily from porcine and murine subjects [34]. Research using excised human intestinal tissue has primarily been limited to studies of drug metabolism and efficacy [35,36]. One of the major inherent problems with the ex vivo approach, particularly in samples obtained from human patients, is the heterogeneity of samples. All individuals have some natural genetic variation, and this difference can be amplified by factors such as gender, age, health status, and environmental influences in sample subjects, making it difficult to compare samples across treatments [37,38,39]. Researchers have also begun using ex vivo methods to examine the effects of nanoparticles on intestinal health. Brun et al. used ex vivo mouse ileum, colon, and gut-associated lymphoid tissue to evaluate translocation and effects of titanium dioxide nanoparticles [40], but most ex vivo intestinal studies have focused on the penetration potential of nanomaterials through the mucus layer [41,42,43]. However, the use of human tissue in evaluating NP is extremely limited, and to date no studies have been published that use human intestinal tissue to investigate toxicity or other adverse impacts of nanoparticle exposure.

The purpose of this study was to evaluate the use of an ex vivo human tissue model to examine sex-dependent effects of AgNP on the human GIT. All tissue samples were excised from human ileal tissue during routine surgical procedures. Integrity of the explant samples was assessed pre-treatment by scanning electron microscopy (SEM). This was done to ensure that the tissue explants retained the complex structure of the intestinal mucosa, including the overlying mucus layer. Tissue explants were transferred to transwell plates and treated with spherical AgNP (10 nm, 20 nm, 75 nm, and 110 nm) at a concentration of 20 µg/mL for 2 and 24 h. Tissue samples were also analyzed using transmission electron microscopy (TEM) after nanoparticle treatment to confirm penetration of nanoparticles into the epithelial tissue. Extracted protein and RNA were then analyzed to evaluate changes in cytokine levels and mRNA expression for genes related to epithelial cell junctions. To reduce the impact of sample heterogeneity, samples were paired, with the response of each explant to nanoparticle treatment measured in comparison to an untreated control explant taken from the same subject.

## 2. Results

### 2.1. Evaluation of Ileal Explants by SEM

Selected intestinal explants were imaged by SEM to evaluate the morphology, appearance, and structural integrity of the ileal biopsy model (Figure 1A). Results of this analysis show that the ileal samples used in this study bore important hallmarks of normal human intestinal tissue. A large number of intestinal villi were observed on the surface of the tissue (Figure 1B). Images taken of the processed tissue explants show that intestinal mucins were also closely associated with the epithelial surface of the ileal samples. Finally, many bacteria were observed in close association with the tissue, indicating the presence of indigenous microflora (Figure 1C).

### 2.2. Evaluation of Ileal Explants by TEM

The purpose for the TEM analysis of the explant samples was two-fold: (1) to evaluate the structural integrity of the tissue sample as a part of the general appraisal of the explant model (pre-treatment) and (2) to confirm penetration of nanoparticles into the tissue (post-treatment). TEM images of the ileal sections before nanoparticle treatment show the intact morphology of the resected section of ileal mucosa which was evidenced by the intact microvilli and underlying architecture of the ileal mucosa, showing tight junctions (zona occludens), intermediate junctions (zona adherens or adheren junctions) and desmosomes (Figure 2A). Upon incubation with 110 nm AgNPs, TEM imaging of the explants show that nanoparticles were present inside the ileal mucosa (Figure 2B,C).

### 2.3. Changes in Cytokine Levels in Human Ileal Tissue after Exposure to AgNPs

Protein homogenates from human intestinal ileal explants taken from five male (Figure 3) and five female subjects (Figure 4) were analyzed for cytokine expression at 2 and 24 h after treatment. Due to the heterogeneity posed by inter individual differences of the human samples, a large variability in normal expression between different cytokines, expression for each group of samples is shown relative to untreated control of same individual. Data for measurement of the cytokine IL-2 is shown only for the 24 h analysis because levels of the cytokine were generally below detection limits at the earlier time point (2 h). Cytokine expression was analyzed by gender to evaluate sex-dependent responses to treatment with AgNPs. The paired analysis revealed that GM-CSF was significantly increased in samples taken from male subjects that were treated with 10 nm and 20 nm AgNPs for 2 h (Figure 3A) compared to untreated controls. Additionally, RANTES was significantly increased in male samples after treatment with 20 nm AgNPs for 24 h (Figure 3B). No significant changes were observed in samples taken from female subjects at the 2 h time point (Figure 4A). However, at the 24 h time point, female samples showed significant decreases in IFN-ɤ and TNF-ɤ after exposure to 10 nm AgNPs and a significant decrease in RANTES after exposure to 20 nm AgNPs (Figure 4B) by the paired t-test, compared to untreated samples. Exposure to AgOAC for 24 h resulted in a significant decrease in the expression of IFN-γ compared to untreated controls (Figure 4B). Several gender-specific trends were observed, although these were not statistically significant. For example, samples taken from female subjects appeared to show greater effects of AgOAC treatment at the 2 h time point. Additionally, male and female subjects appeared to have different responses to treatment with the 10 nm AgNPs at the 2 h time point, as expression of IL-6, IL-8, IL-10, TNF-α, and RANTES were increased in samples excised from male subjects, but decreased or unchanged in samples taken from female subjects.

A combined paired analysis of all samples (Appendix A
Figure A1A) showed that after 2 h the samples treated with 110 nm AgNPs exhibited a significant increase in the cytokine IL-1b, while the samples treated with 20 nm and 75 nm AgNPs exhibited a significant increase in the cytokine GM-CSF. IL-1b and IL-8 levels were more than 2-fold higher than controls in samples treated with 20 nm AgNPs for 2 h, but this result was not significant. More significant effects were detected at the 24-h time point, as exposure to the 10 nm AgNPs resulted in significantly decreased levels of IL-4, IL-10, and IFN-ɤ and exposure to the 75 nm AgNPs resulted in a decrease in IL-2, IL-4, IL-6, IFN-ɤ, and an increase in RANTES (Appendix A, Figure A1B). Decreased expression of IL-4, IFN-ɤ, and TNF-α was also seen in response to treatment with AgOAC for 24 h.

### 2.4. Changes in mRNA Expression of Epithelial Permeability-Related Genes in Human Ileal Tissue after 24-h Exposure to AgNPs

mRNA expression of most cell junction genes was altered in response to AgNPs’ exposure at the 24 h time point, albeit with major differences between male (n = 3) and female (n = 3) samples (Figure 5). Size-dependent differential effects of AgNPs in male and female samples were noted for genes related to epithelial permeability. Smaller size AgNPs (10 nm) perturbed more genes in male ileal mucosa than in females. However, this trend was reversed during exposure to 20 nm AgNPs or AgOAC, revealing a larger number of upregulated genes in female samples compared to male samples. Moreover, fewer genes were downregulated with a lower magnitude of downregulation in the female samples.

As an example, the relative expression of a selected gene compared to the untreated control from each of the six major cell junction families (focal adhesions, gap junctions, adherens junctions, tight junctions, desmosomes, and hemidesmosomes) after nanoparticle treatment is shown in Figure 6. Overall, male samples seemed to be more affected by AgNP exposure than female samples. Interestingly, male samples also largely experienced upregulation in cell junction genes, while female samples exhibited a much more varied response, with a mix of both upregulation and downregulation. Caveolin 1 (*CAV1*) was upregulated in male samples treated with AgNPs but downregulated in female samples treated with AgNPs. In fact, female samples treated with 20 nm AgNPs showed a significant (more than 20-fold; *p*-value = 0.01) downregulation in *CAV1*. Another noteworthy gene showing differential expression was Claudin 17 (*CLDN17*), which was upregulated in males and greatly downregulated in females.

## 3. Discussion

The human ileal samples used in this study showed differential cytokine responses to AgNP treatment dependent on duration of exposure, particle size, and subject sex. At the 2 h time point, all significant cytokine responses were increased compared to control tissue response, whereas at 24 h most significant responses were decreased compared to the control response. Silver nanoparticles of different sizes also evoked different responses, with the smaller particles (10 nm, 20 nm, and 75 nm) generally associated with a greater effect than the 110 nm AgNPs. This could have been due to the uptake mechanism, as the uptake of particles greater than 100 nm has been shown to be primarily through M-cells, while those less than 100 nm generally enter through M-cells, enterocytes, and goblet cells [40]. Sex-related differences in tissue response were also observed. For example, the cytokine RANTES was significantly increased in samples taken from male subjects treated with 20 nm AgNPs for 24 h, and significantly decreased in samples taken from female subjects. Expression of the RANTES protein, also known as Chemokine ligand 5 (CCL5), attracts monocytes and T-cells, suggesting that explants taken from male samples might have had a greater tendency to invoke an inflammatory response upon exposure to 20 nm AgNPs. No significant effects were observed at the 2 h time point after treatment with AgOAC, which was used as a positive control for silver ion exposure in this study. However, significant decreases in IL-4 and IFN-γ were seen in samples treated with AgOAC at 24 h, which correlated with similar decreases in samples treated with 10 nm and 75 nm AgNPs. This indicates that the silver ions released from these particles may be responsible for the observed effect.

Sex-dependent changes were also observed in the mRNA expression of cell junction related genes after treatment with AgNPs. Significant changes in the inflammatory response, as measured by cytokine expression, were only observed for smaller AgNPs (10 nm and 20 nm), thus we selected these samples for further investigation into mRNA expression of intestinal permeability-related genes. It is likely that smaller AgNPs (10 nm and 20 nm) were able to penetrate through the paracellular spaces in the intestinal epithelium and impact cellular structures involved in epithelial permeability, such as cell junctions. Cell junction structures, including tight junctions and focal adhesions, have been known to play an important role in cell-cell communication and adhesion [44,45,46]. It is important to note that the smallest AgNPs (10 nm) affected cell junction gene expression most adversely as was also shown in our previous study when polarized intestinal epithelial cells were used [22]. As hormonal activity was still present in ex vivo tissue, it is possible that the observed difference between male and female samples could be attributed to differences in endocrine levels. In fact, there have been relationships drawn between specific genes and sex hormones in previous studies conducted in rodents and zebrafish [47,48]. One gene, *CAV1*, that was significantly downregulated in this study in female samples exposed to 20 nm AgNPs, has been linked previously to estrogen levels in rats [49,50]. Given that *CAV1* has been shown to be protective against intestinal inflammation in some circumstances, this suggests that females could be more at risk of an inflammatory response after treatment with 10 nm and 20 nm AgNPs [51,52].

A number of studies in the past decade have raised concerns about the reliability of methods used to model and test intestinal mucosa [23,31,32,34,53]. One of the most common strategies is the in vitro use of intestinal cell lines, but this approach has been shown to have significant drawbacks, including technical limitations and difficulty in correlating results with in vivo data obtained from animal models [23,34]. Most importantly, data interpretation from the in vitro cell culture models is applied globally, that is, without consideration of sex-dependent responses. Our previous work in T84 cell lines also showed a greater effect with smaller (10 nm) AgNPs [22]. Interestingly, in the in vitro paper, 20 nm AgNPs treatment resulted in significantly decreased expression of the DSC3 gene, involved in cell-cell adhesion, whereas little to no change was observed in either male or female samples in this work. Our preceding studies in rats also showed that AgNP mediated response in a size- and sex-dependent manner [22,48]. Both studies, which were examinations of the intestinal effects of 13-week AgNP administration in rats, found that the greatest effects were seen after administration of 10 nm AgNPs. Additionally, Williams et al. found that exposure to 10 nm and 75 nm AgNPs resulted primarily in down-regulation of host genes, particularly genes related to microbial recognition and T-cell function. This observation, combined with the general depression of cytokine expression observed after exposure to AgNPs in this study, could indicate that AgNPs may depress immune function or activation in the intestinal tract. In contrast, the study published by Orr et al. found an increased level of TNF-α in female rats treated with 10 nm AgNPs and a dramatically different pattern of gene expression related to cellular adhesion and permeability. This discrepancy could be fundamentally host-dependent (rat vs. human) or due to the considerable difference in the length of AgNP exposure (13 weeks vs. 24 h).

In this study, we used a translational model from rodent to human by using an ex vivo approach. This ex vivo model, which utilizes tissue explants obtained from routine human surgical procedures, has generally been a less common model of evaluating the intestinal epithelium, but has many benefits as presented in the current study. For example, tissue explants contain a full cross-section of the intestinal wall. SEM and TEM analyses of the ileal samples used in our study demonstrate that the explants are intact and contain mucus layer, intestinal villi, underlying submucosa, and mucosa-associated microbial population; and suggest that cellular polarity is maintained. The presence of this architectural layering is important for an accurate assessment of the impact of a xenobiotic agent, as the first contact of such test agents/substances entering the intestinal environment is not with epithelial cells, but with the overlying mucus layer [23]. As this study was intended to be an initial appraisal of the ex vivo model for evaluating the impact of nanomaterials in the GIT, no in-depth histopathological survey was made of the treated tissue samples. However, this type of work will be critical in validating this model for future use and should include an in-depth analysis of visible markers of inflammation and disrupted barrier function. The primary limitation of the ex vivo model is the short duration available for testing [36]. As a result, tissue explants used in this study were not evaluated past 24 h of treatment. However, most of the xenobiotic or test agents that pass through the intestinal tract do not remain in the ileum longer than the defined period of treatment conducted in the present study.

Within the intestinal mucosa, changes in intestinal gene expression and protein secretion reflect a complex network of inter-dependent relationships that are vulnerable to xenobiotic perturbations, despite multiple chemokine ligand-receptor redundancies. The primary checkpoint in the protection of this system is the intestinal epithelium. Silver (in nanoparticle form or in ion form) can penetrate this defensive layer in several ways, raising questions about the safety of some consumer goods containing AgNPs. This study used ex vivo intestinal ileal explants excised from human subjects as a model to examine the inflammatory effects of nanomaterials on the intestinal tract. Limited availability of healthy excised intestinal tissue from humans could be a challenge in increasing the number of samples and, therefore, the repeatability of the tests and the variability of research cohort. Nonetheless, a major benefit of this approach is the ability to obtain multiple biopsy samples from each tissue sample and to retain one as an untreated control for each time point. This allowed paired comparison of subject tissues with and without AgNP treatment during analysis and helped to mitigate one of the major concerns of working with ex vivo explants, which is sample variability [23]. Our study also emphasizes that researchers should exercise caution when inferring results obtained from pooled cohorts, more specifically averaging male and female data, as it may lead to misinterpretation of the real outcome. This ex vivo model could also serve as a useful tool for evaluating intestinal mucosa-associated microbial population changes during exposure to exogenous compounds, a strategy currently under investigation in our laboratory. We believe this ex vivo model may be a valuable tool in future research, particularly for evaluating more subtle changes in inflammatory response, immunological changes, and changes in microbial population resulting from xenobiotic exposure.

## 4. Materials and Methods

### 4.1. Characterization of Nanoparticles

All nanoparticles used in this study were characterized as a part of previous work [17,22,54], and detailed information, including dynamic light scattering, inductively coupled plasma-mass spectrometry, and zeta-potential analysis for these materials is presented elsewhere [54]. Briefly, spherical AgNPs in sizes 10, 20, 75, and 110 nm were obtained from NanoComposix (San Diego, CA, USA) in BioPure formations. AgNPs were suspended in a 2 mM sodium citrate solution. Nanoparticles were checked for sterility, endotoxin contamination, and protein corona formation as previously described [22]

### 4.2. Procurement of Excised Intestinal Tissues

This study was conducted under the approved protocol for obtaining intestinal tissue samples (terminal ileum) from the Cooperative Human Tissue Network (CHTN) located at the University of Alabama at Birmingham. The CHTN coordinates with other centers to provide high-quality tissue specimens to researchers. The use of these tissue samples was approved by US Food and Drug Administration’s Research Involving Human Subject Committee. The criteria for the inclusion and/or exclusion of the human subjects was that the samples should be from de-identified subjects who are above age of 18 years and do not have inflammatory bowel disease or pathological condition involving entire bowel. All specimens were human terminal ileum taken from subjects as a part of routine surgical procedures. Details of these specimens and the assays in which they were used are provided in Appendix A
Table A1. All tissue culture items were procured from ThermoFisher Scientific, Waltham, MA, USA. Samples were shipped overnight on ice in nutritive transport media (RPMI 1690 with GlutaMAX, supplemented with pen/strep) from the coordinating Center to the National Center for Toxicological Research in Jefferson, AR. All specimens used in this study were found to be normal tissue by post-surgical pathological examination. All patient information was protected and the only information available for each sample was age, race, and sex (Appendix A
Table A1).

### 4.3. Treatment of Ileal Tissue

Once received, all patient specimens were processed according to strict handling procedures. For each set of experiments, a positive control (tissue exposed to silver acetate; AgOAC) and a negative control (tissue without nanoparticle) were maintained separately as described below. Ileal tissue specimens were removed from transport media and washed with RPMI media, then placed onto dental wax and cut using a 6 mm biopsy punch (ThermoFisher Scientific, Waltham, MA, USA). Individual tissue biopsy punches were placed into 6-mm polyester transwell inserts (Corning, Corning, NY, USA), one biopsy punch per transwell, with the mucosal side facing up. Culture media, composed of RPMI 1690 with GlutaMAX, supplemented with 10% fetal bovine serum (FBS), pen/strep, and fungizone, was then added to the apical and basal transwell compartments. Finally, AgNPs or a positive control (AgOAC) were added to the apical compartments of each transwell at final concentrations of 20 µg/mL. The same volume of media alone was used as the negative control. Tissue punches treated with nanoparticles or controls were incubated with 5% CO_2_ at 37 °C for 2 and 24 h. Samples were collected for electron microscopy, extraction of protein or mRNA as described below.

### 4.4. Examination of Ileal Tissue by Electron Microscopy

To determine if the ileal samples maintained their integrity during sample transport, some ileal specimens (obtained during the initiation of this study) were processed and treated as previously described for evaluation by SEM and TEM. The biopsy punches were placed in 24-well plates and treated as described earlier. At the designated time point, individual biopsy punches were transferred to 1 mL 3% glutaraldehyde (Electron Microscopy Sciences, Hatfield, PA, USA). Samples were then processed in the NanoCore facility at the National Center for Toxicological Research for evaluation by SEM and TEM as described earlier with FE-SEM (Zeiss Merlin, Oerzen, Germany) and JEM-2100 200 keV (JEOL USA, Inc., Peabody, MA, USA) instruments [55].

### 4.5. Extraction of Protein and RNA from Ileal Tissue

At the designated time point, tissue punches were removed from transwell plates and processed to extract RNA and protein using the mirVana PARIS RNA and Protein Purification Kit (ThermoFisher Scientific, Waltham, MA, USA). Briefly, tissue punches were minced into 1-mm segments using a sterile scalpel and then segments were placed into ice-cold Dissociation Buffer and homogenized using a pellet pestle motor and a Micro Tissue Homogenizer set (ThermoFisher Scientific). The remainder of the protein extraction was then performed according to the manufacturer’s instructions. RNA extraction from the samples was completed as described earlier [56]. Protein and RNA samples were stored at −80 °C until further use.

### 4.6. Measurement of Cytokine Levels in Ileal Tissue

Cytokine levels in ileal tissue samples were measured using the Bio-Plex Multiplex Immunoassay System (Bio-Rad, Hercules, CA, USA), which uses a bead-based multiplex system to quantify specific target proteins. Protein extracts were thawed on ice and protein concentration was measured using Bradford reagent (Bio-Rad). Samples were adjusted to 500 µg/mL total protein and plates were prepared for analysis using protein extracts (two replicates per individual; with a total of five male and five female), reagents, and instructions from two different Bio-Plex Pro Human Cytokine kits: an 8-plex assay for the measurement of the cytokines IL-1β, IL-2, IL-4, IL-6, IL-8, IL-10, GM-CSF, IFN-γ, and TNF-α and a custom-created assay designed for the measurement of the cytokines IL-12 (p70), Basic FGF, IP-10, RANTES, and VEGF. The 8-plex assay was chosen as a general panel of cytokines involved in the intestinal immune response, including cytokines involved in pro- and anti-inflammatory responses and T-cell signaling. Additional cytokines specifically related to conditions of chronic intestinal inflammation were custom designed. Plates were read on a Bioplex 200 instrument using Bio-Plex Manager software (Bio-Rad). Raw data was exported and analyzed using the Bio-Plex Data Pro Software (Version 6.2, Bio-Rad, Hercules, CA, USA). Statistical analysis was performed using a paired t-test, with each nanoparticle-treated explant paired to its untreated control. Significance was designated as a *p*-value < 0.05.

### 4.7. mRNA Gene Expression Analysis

Quantity and quality of RNA extracted from ileal tissue samples at the 24-h time point were analyzed using a NanoDrop^®^ ND-1000 spectrophotometer (NanoDrop, Wilmington, DE, USA). The Turbo DNA-free kit (Life Technologies, Carlsbad, CA, USA) was used to remove contaminating DNA. Treated RNA was re-quantified and reverse transcribed using the Two-Step Reverse Transcription TaqMan kit (Applied Biosystems, Foster City, CA, USA) to produce cDNA. Sample cDNA was run on a RT2 Profiler PCR Array Human Cell Junction Pathway Finder (Qiagen, Valencia, CA, USA). Eighty-four unique genes related to cell junctions and intestinal permeability were examined including adherens junctions, focal adhesions, tight junctions, gap junctions, desmosomes, and hemidesmosomes. Plates were analyzed using an ABI 7500 Real-Time PCR system (Life Technologies). Data was analyzed in the Qiagen Data Analysis Center using glyceraldehyde-3 phosphate dehydrogenase (GAPDH) as the housekeeping gene for normalization. The qPCR data was analyzed statistically using a student’s t-test. Significance was designated as a *p*-value < 0.05.

## 5. Conclusions

Several nanoparticles, including silver nanoparticles (AgNPs) are used as health supplements, antimicrobials and additives to increase the shelf life of the perishable products. This study utilized human intestinal explants to assess the effects of AgNPs on the gastrointestinal tract (GIT). The outcome of this study reveals that the human ex vivo tissue model can distinguish sex-dependent effects of AgNPs on the human GIT in terms of intestinal immune response and epithelial barrier function; thus could serve as a non-animal translational model that is more relevant to human population.

## Figures and Tables

**Figure 1 ijms-22-00009-f001:**
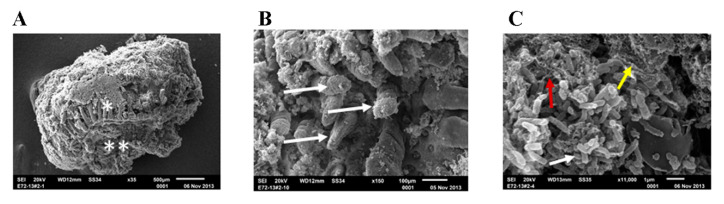
Scanning Electron Micrographs (SEM) of ex vivo ileal tissue before treatment. Untreated ileal tissue showing the integrated architecture of the mucosa ready to be examined by the SEM after 24 h incubation in transwells; * indicates luminal surface and ** indicates serosa surface (**A**). Under higher magnification ileal tissue shows intact villi (shown by white arrow in (**B**), mucosa-associated bacteria (shown by white arrow), fibrin (shown by red arrow) and biofilm (shown by white, red and yellow arrow, respectively in (**C**).

**Figure 2 ijms-22-00009-f002:**
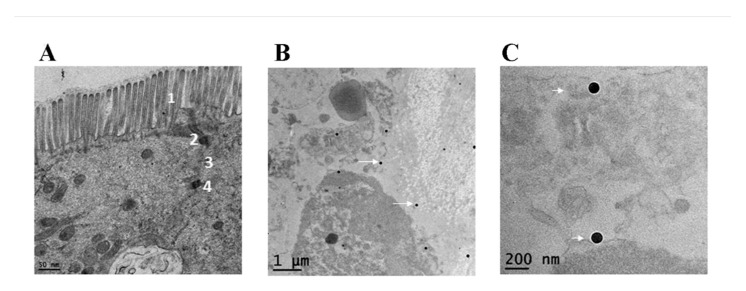
Transmission Electron Micrographs (TEM) of ex vivo ileal tissue. The TEM of ileal tissue Scheme 1. as well as tight junction (indicated as 2), intermediate junctions (indicated as 3) and desmosomes (indicated as 4) in (**A**). After 24 h of antimicrobial silver nanoparticles (AgNP) treatment, (**B**,**C**) show presence of 110 nm AgNPs inside the ileal mucosa as indicated by white arrows.

**Figure 3 ijms-22-00009-f003:**
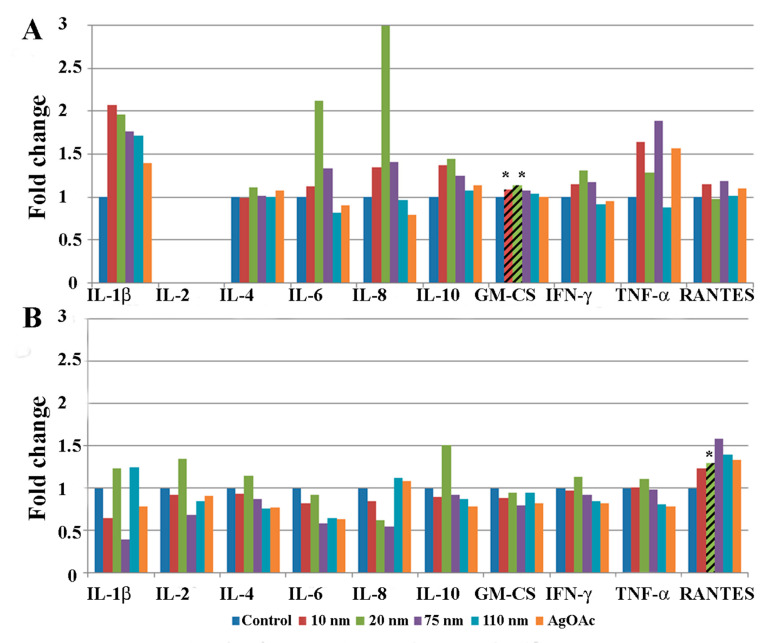
Effect of AgNP exposure on the ileal mucosa-associated immune response in samples taken from male subjects. Gut-associated mucosal chemokine/cytokines levels were evaluated in the tissue lysate after 2 h (**A**) or 24 h (**B**) of AgNP exposure in male samples. Results shown are changes in chemokine/cytokine levels relative to untreated control biopsy punch (n = 5). Marked bars denote statistical significance by paired t-test, * indicates *p* < 0.05 compared to untreated controls.

**Figure 4 ijms-22-00009-f004:**
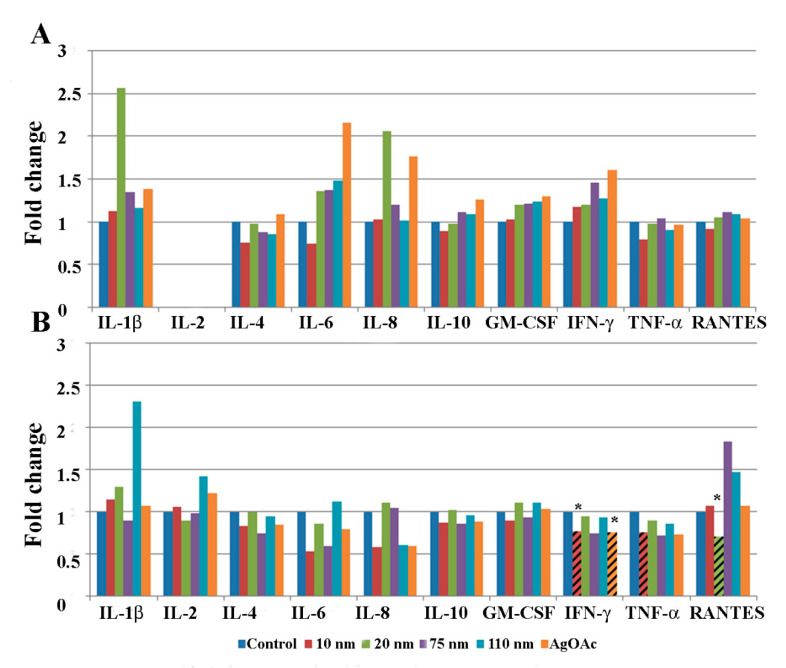
Effect of AgNP exposure on the ileal mucosa-associated immune response in samples taken from female samples. Gut-associated mucosal chemokine/cytokines levels were evaluated in the tissue lysate after 2 h (**A**) or 24 h (**B**) of AgNP exposure in female samples. Results shown are changes in chemokine/cytokine levels relative to untreated control biopsy punch (n = 5). Marked bars denote statistical significance by paired t-test, * indicates *p* < 0.05 compared to untreated controls.

**Figure 5 ijms-22-00009-f005:**
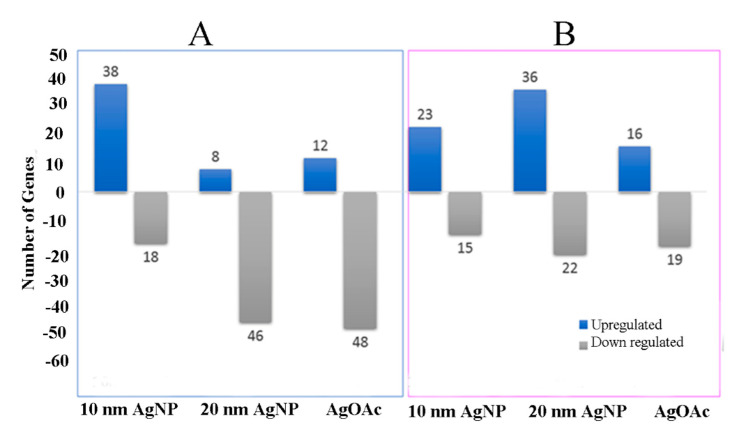
mRNA expression for genes involved in mucosal permeability. Real-time PCR analysis of mRNA expression for genes involved in the mucosal permeability for male and female samples are shown (**A**) and (**B**), respectively. *Y*-axis shows the number of genes differentially expressed (as compared to control) during the treatment of ex vivo tissue with AgNP or AgOAC for 24 h.

**Figure 6 ijms-22-00009-f006:**
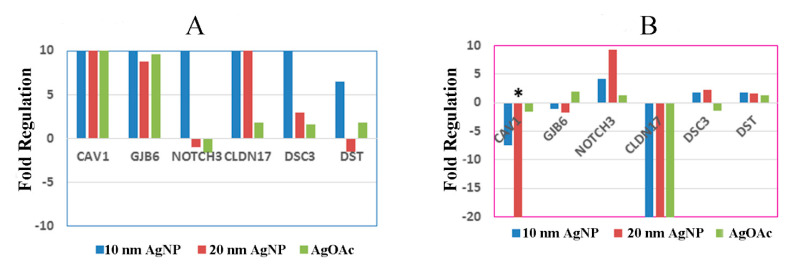
Effect of AgNP exposure on the differential expression of mRNA for ileal mucosa-permeability related genes in male and female. Fold regulation of cell junction genes in male and female ((**A**) and (**B**), respectively) ex vivo tissues during treatment with AgNP and AgOAC. * indicates *p* < 0.05 compared to untreated controls.

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
