# Peer review of "Human Intestinal Tissue Explant Exposure to Silver Nanoparticles Reveals Sex Dependent Alterations in Inflammatory Responses and Epithelial Cell Permeability"

_ijms, 2020, doi:10.3390/ijms22010009_

Round 1
Reviewer 1 Report
The study is interesting, but limited and raises various questions.
Sampling is small and heterogenous in different ages and biological conditions.
What are the physiopathological conditions of the subjects ?
Criteria for inclusion and/or exclusion should be considered.
Author Response
On behalf of all the authors, we highly appreciate positive feedback provided by reviewer. We have edited the manuscript to add more clarity. We hope that the revised manuscript will be acceptable to the reviewer. Please see below the responses for comments.
Comment: The study is interesting, but limited and raises various questions. Sampling is small and heterogenous in different ages and biological conditions.
Response: This study was conducted as a proof of concept to use ex vivo excised tissue from different age groups to assess the possibility of developing a non-animal translational model that is relevant to human. Thus, we included subject from diverse age group (above 18 years). Condition of all tissues were healthy. Despite the study participants were heterogenous in different ages, the outcome of the study showed significant differences in the endpoints. Moreover it matched our earlier published results in the rodent model.
Comment: What are the physiopathological conditions of the subjects ?
Response: The physiopathological condition of the tissues were healthy, as our inclusion criteria specified that "the participant should not have Inflammatory bowel disease or pathological condition involving entire bowel." We have mentioned this in our revised manuscript.
Comment: Criteria for inclusion and/or exclusion should be considered.
Response: Criteria for inclusion and/or exclusion were as follow:
De-identified subjects who are above age of 18 years and do not have Inflammatory bowel disease or pathological condition involving entire bowel can participate.
Reviewer 2 Report
The authors sought to evaluate the use of an ex vivo human tissue model to examine sex-dependent effects of AgNP on the human GIT. I felt that a lot of effort was contributed to achieve this goal, and this manuscript is clear logic and well-written. However, it is mandatory to explain/correct the manuscript in some points.
- How to calculate p-value with only 2 replicates (line 390)?
- Error bars should be provided in Fig. 3 and 4.
Author Response
On behalf of all the authors, we highly appreciate positive feedback provided by reviewer. We have edited the manuscript to add more clarity. Please see below the responses for comments . We hope that the revised manuscript will be acceptable to the reviewer.
Comment 1. How to calculate p-value with only 2 replicates (line 390)?
Response: Here two replicates were from each individual (technical replicate); however the p value is calculated by taking the values from all five individuals (five biological replicates).
Comment 2. Error bars should be provided in Fig. 3 and 4.
Response: We highly appreciate reviewer’s comment. Results shown are changes in chemokine/cytokine levels relative to untreated control biopsy punch from same individual (please see that all controls are 1). There are no error bars on these figures because the graphs represent fold change compared to untreated control, not measured values, therefore standard deviation as error bar is not recommended on such data. The y axis should be fold change, we have now changed it in the figure.
Round 2
Reviewer 1 Report
The authors provided appropriate and satisfactory reply.
The manuscript is convincing.
Reviewer 2 Report
All concerns were addressed.